# Impact of human papillomavirus age-related prevalence and vaccination levels on interpretation of cervical screening modalities: a modelling study

David Robert Grimes  [1,2]

¹School of Medicine, Trinity College Dublin, Dublin, Ireland
²School of Physical Sciences, Dublin City University, Dublin, Ireland

**Correspondence to**
Dr David Robert Grimes; davidrobert.grimes@tcd.ie

## ABSTRACT

**Objective** Cervical screening is a life-saving intervention, which reduces the incidence of and mortality from cervical cancer in the population. Human papillomavirus (HPV) based screening modalities hold unique promise in improving screening accuracy. HPV prevalence varies markedly by age, as does resultant cervical intraepithelial neoplasia (CIN), with higher rates recorded in younger women. With the advent of effective vaccination for HPV drastically reducing prevalence of both HPV and CIN, it is critical to model how the accuracy of different screening approaches varies with age cohort and vaccination status. This work establishes a model for the age-specific prevalence of HPV factoring in vaccine coverage and predicts how the accuracy of common screening modalities is affected by age profile and vaccine uptake.

**Design** Modelling study of HPV infection rates by age, ascertained from European cohorts prior to the introduction of vaccination. Reductions in HPV due to vaccination were estimated from the bounds predicted from multiple modelling studies, yielding a model for age-varying HPV and CIN grades 2 and above (CIN2+) prevalence.

**Setting** Performance of both conventional liquid-based cytology (LBC) screening and HPV screening with LBC reflex (HPV reflex) was estimated under different simulated age cohorts and vaccination levels.

**Participants** Simulated populations of varying age and vaccination status.

**Results** HPV-reflex modalities consistently result in much lower incidence of false positives than LBC testing, with an accuracy that improves even as HPV and CIN2+ rates decline.

**Conclusions** HPV-reflex tests outperform LBC tests across all age profiles, resulting in greater test accuracy. This improvement is especially pronounced as HPV infection rates fall and suggests HPV-reflex modalities are robust to future changes in the epidemiology of HPV.

## INTRODUCTION

Cervical cancer screening is a powerful life-saving intervention, associated with significant reductions in the mortality and cancer burden.[1–3] For squamous cell carcinoma in women older than 25 years, national screening paradigms have resulted in an estimated 80% reduction in mortality. Human

---

## STRENGTHS AND LIMITATIONS OF THIS STUDY

⇒ Extends modelling approaches to screening outcomes by incorporating the projected human papillomavirus (HPV) prevalence and vaccine uptake levels of age-varying cohort.

⇒ Provides a novel methodology for ascertaining how screening modalities will perform under likely future scenarios.

⇒ HPV prevalence estimates are limited by assumptions efficacy and uptake of vaccination, and this can be varied with the model presented.

---

papillomavirus (HPV) is responsible for most of these cancers, and the advent of vaccination against its oncogenic strains has already had transformative impact on both rates of HPV and cervical intraepithelial neoplasia (CIN).[2]

Alongside conventional liquid-based cytology (LBC) modalities for screening for CIN grades 2 and higher (CIN2+), the evolution of screening modalities to include HPV-reflex method, which triage cases by HPV infection status have shown great promise.[4–6] This promise, however, requires careful interpretation—HPV testing is often deemed to have higher sensitivity than conventional LBC approaches, but there are nuances which demand careful consideration. The sensitivity of HPV tests is not a direct proxy to LBC sensitivity, as they measure different things. While LBC tests directly measure the presence of CIN2+, HPV tests themselves only detect the presence of high-risk strains of HPV. In younger women especially, there is high transient incidence of HPV, and using HPV infection alone as a screening modality would result in an unacceptably high false detection rate of CIN2+. False positive results not only induce anxiety[7 8] in patients but also lead to potential overtreatment, with the potential to overwhelm colposcopy services unless these are robustly designed.[3 9]

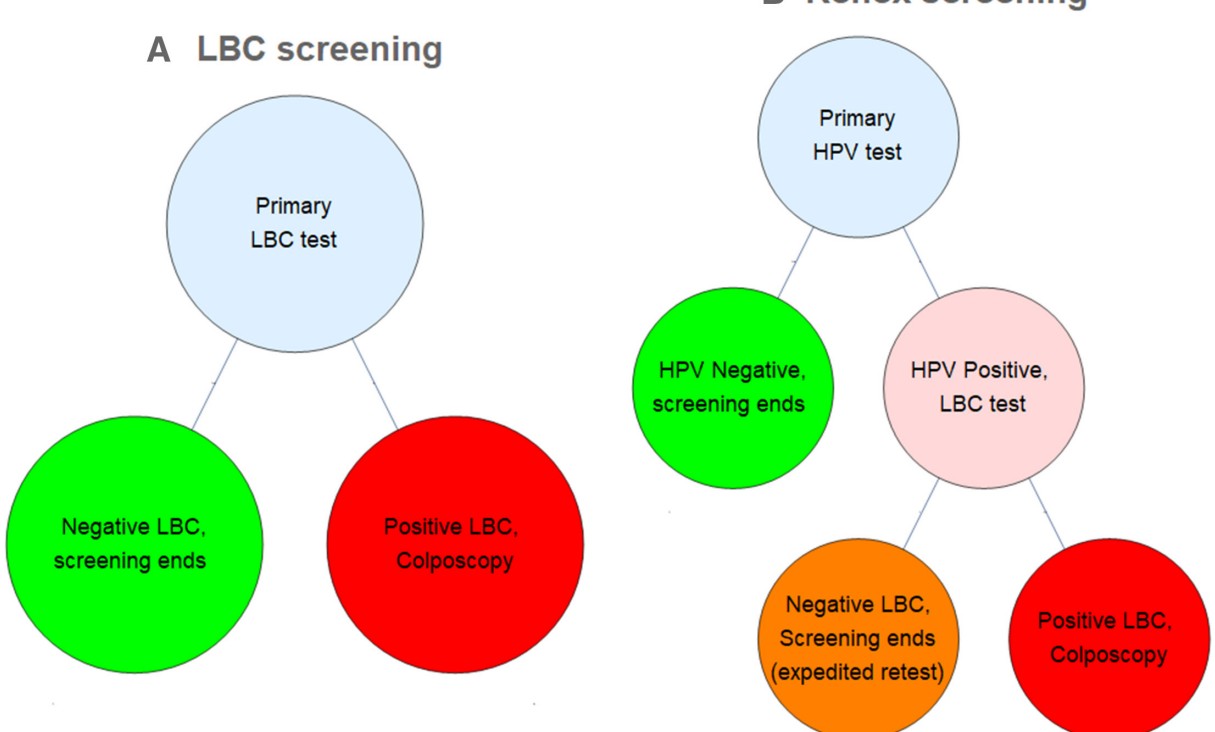

**Figure 1** Clinical patient flow for (A) LBC screening and (B) HPV reflex screening. HPV positive, LBC negative results are frequently selected for an expedited retest outside of the normal screening interval. HPV, human papillomavirus; LBC, liquid-based cytology.

Accordingly, HPV triage approaches are typically implemented, where patients testing positive for high-risk HPV receive a reflex LBC, while those testing negative do not. A recent analysis[3] demonstrated that this reflex approach has in fact a slightly lower effective sensitivity for CIN2+ (0.68, 95% CI 0.60 to 0.75) than LBC testing alone (0.76, 95% CI 0.67 to 0.83) in a typical population, though this seeming disadvantage is typically mitigated by expedited retesting of HPV positive results before the next screening cycle. The chief advantage of HPV-reflex tests (or LBC tests with HPV reflex) is that they hugely decrease the false positive rate, with a reflex test yielding 9.6 false positives per 1000 women (95% CI 9.4 to 9.8) compared with LBC testing with 95.1 false positives per 1000 women (95% CI 93.7 to 97.0), decreasing false positive rates by approximately 10-fold and reducing over-treatment. Simplified clinical patient flow for both LBC and HPV-reflex modalities is illustrated in figure 1.

Yet to date, this work has been largely considered average HPV infection rates and CIN2+ prevalence averaged across entire populations. HPV infection rates, however, differ by both age cohort[10] and vaccination status.[11] Younger women tend to have higher rates of both HPV infection and CIN2+. This in turn has implications for current and future screening interpretations and planning with differing cohorts. CIN2+ prevalence is directly proportional to HPV infection[12] and will change with as infection rates change either due to ageing cohorts or increased vaccine uptake. A simple dynamic model to account for this would allow accurate determination of screening performance under realistic scenarios and facilitate the planning of future screening modalities. This work accordingly establishes a model to account HPV infection rates, and CIN2+ prevalence with age cohorts and vaccine uptake, ascertaining test accuracy under all possible future scenarios.

## METHODS
### Modelling varying prevalence of HPV and CIN2+
HPV infection rates vary markedly by country and cohort, and even detection methodology. The evidence to date strongly suggests that HPV infection peaks before 30, though slightly different trends emerge in various data sets. For the purposes of this investigation, analysis was restricted to sets prior to widespread coverage of the HPV vaccine. On current estimates from the USA and European data, we expect a global average of approximately 8.4% of the total population to have active HPV infection at the time of screening.[3] This figure pertains to the total prevalence across the population, and the caveats are elucidated in the discussion.

For this work, it was crucial to further stratify this figure by age cohort. To derive an age-specific HPV prevalence rate, we considered the 2008 review by Smith *et al*[10] of the literature on HPV infection rates throughout the world. This review included various data sources, including these stemming from sexual infection clinics

or specific high-risk populations. As these results are likely to produce high (or in some cases low) outliers which can skew analysis, we confined ourselves only to the subset of studies solely from general European screening programmes. Furthermore, this analysis only considered studies with some degree of age stratification. This resulted in 12 applicable studies.[13–24] These studies were undertaken in various cities throughout Belgium, Denmark, France, Italy, The Netherlands and the UK between 1992 and 2005, with samples ranging from 323 women to 7932 women (mean: 2878 women). Data from these studies were then taken and weighted according to the number of study participants and smoothed, so that an age-varying phenomenological model describing HPV prevalence at age in years $a$ could be ascertained with 95% CIs, denoted $h_u(a)$. Details of these studies, participants, settings and HPV subtypes tested for are given in online supplemental material.

Crucially, these estimates stem from before the advent of HPV vaccination and accordingly map to a scenario of 0% vaccine coverage. This is a useful baseline on which we can model the impact of vaccination on test performance. The impact of HPV vaccination on prevalence rates was derived from 29 previously published modelling studies,[8] yielding an estimate for $V$, the reduction factor for HPV prevalence from given level of vaccination, as previously published.[3 11] The time-varying HPV prevalence for a given level of vaccination is thus $h(a) = (1 - V) h_u(a)$, with the mathematical justification for this simplification given in the supporting mathematical appendix. For brevity, results in this work consider the impact of vaccine rates of 0% (completely unvaccinated cohorts) and 80% (highly vaccinated cohorts) with mid-level vaccination rates of 40% given in online supplemental material. The mean value for $h(a)$ across the entire span of screening ages is given by

$$\overline{h} = \frac{1}{a_e - a_s} \int_{a_s}^{a_e} h(a)\, da.$$

Taking the upper bound of screening ages to be $a_e = 65$ years and the lower bound to be $a_s = 20$ years, the mean HPV prevalence in the cohort can be ascertained from the form of the phenomenological function. The CIN2+ prevalence is proportional to the rates of HPV infection, with a justifying argument for this given in the mathematical online supplemental material. It follows then that a simple scaling function should be predictive of prevalence rates for CIN2+. If the known population average of CIN2+is $\overline{p}$, then age/vaccine varying CIN2+prevalence is accordingly given by

$$p(a) = \left(\frac{\overline{p}}{\overline{h}}\right) h(a).$$

### Interpretation of screening results with varying HPV prevalence and vaccine uptake

With age and vaccination-specific CIN2+ prevalence known, one can explicitly calculate false positive rates,

false negative rates and overall correct identifications using previously published methods. For LBC, the false positive ($fp_l$), false negative ($fn_l$) and overall correct identification rates ($t_l$) are related to CIN2+ prevalence and the sensitivity ($s_{nl}$) and specificity ($s_{pl}$) of LBC testing, given, respectively, by the relationships

$$fp_l(a) = (1 - p(a))(1 - s_{pl})$$
$$fn_l(a) = p(a)(1 - s_{nl})$$
$$t_l(a) = 1 - (fp_l(a) + fn_l(a)).$$

For HPV reflex testing, false positive rates ($fp_r$), false negative rates ($fn_r$) and overall correct identification rate ($t_r$) depend additionally on HPV prevalence, the sensitivity of HPV testing ($s_{nh}$), the specificity of HPV testing ($s_{ph}$) and the fraction of CIN2+ cases associated with detectable HPV, $v$. These equations are derived from those previously published[3] and are, respectively, as follows:

$$fp_r(a) = \begin{aligned} &(1 - s_{pl})(s_{nh}(h(a) - p(a)v)\\ &+ (1 - s_{ph})(1 - h(a) - p(a)(1 - v)) \end{aligned}$$
$$fn_r(a) = vp(a)(1 - s_{nl}s_{nh}) + p(a)(1 - v)(1 - s_{nl}(1 - s_{ph}))$$
$$t_r(a) = 1 - (fp_l(a) + fn_l(a)).$$

Literature values used for all modelling simulations are given in table 1.

### Patient and public involvement

None. It was not appropriate or possible to involve patients or the public in the design, or conduct, or reporting, or dissemination plans of this research.

## RESULTS
### Modelling varying prevalence of HPV and CIN2+

Age-related HPV prevalence from 12 included studies were found to approximately obey an exponential decay type-model from ages 20–65 of the form $A \exp(-\beta t)$ as illustrated in figure 1 ($A = 35.81 \pm 4.91$, $\beta = 0.0336 \pm 0.0045\, yr^{-1}$, 95% CI). This envelope with weighed study data points is shown in figure 1. The estimate for mean HPV infection rate in unvaccinated populations across all age ranges was $\overline{h} = 9.4 \pm 2.3\,\%$, in good agreement with literature estimate of 8.4%, and with population-wide European estimates of 9.8%[25] (95% CI 9.2% to 10.0%) and from the ATHENA study[26] (10.5%, 95% CI 10.3% to 10.7%). CIN2+ rates by age cohort from inferred from this as per the model outline.

### Interpretation of screening results with varying HPV prevalence and vaccine uptake

Figure 2 depicts how the accuracy of LBC and HPV-reflex modalities (in terms of true results, false positives and false negatives) change with age in both unvaccinated and 80% vaccinated populations. HPV reflex modalities consistently outperform LBC modalities for all age cohorts and vaccine uptake levels. Table 2 depicts a selection of results for varying age profiles and vaccine

**Table 1** Simulation values and source

| Parameter | Values (95% CI where available) |
|---|---|
| Average prevalence of CIN2/3 in full population ($\bar{p}$) | 0.02[6] |
| Sensitivity of LBC test to CIN2+ ($s_{nl}$) | 0.76±0.04[6] |
| Specificity of LBC test to CIN2+ ($s_{pl}$) | 0.903±0.001[6] |
| Sensitivity of HPV test to high-risk HPV ($s_{nh}$) | 0.947 (–)[3] |
| Specificity of HPV test to high-risk HPV ($s_{ph}$) | 0.96±0.001[6] |
| CIN2/3 attributable to testable hr-HPV ($v$) | 0.95 (–)[3] |
| Reduction in HPV at 40% vaccine coverage ($V_{40}$) | 0.53±0.06[3 11] |
| Reduction in HPV at 80% vaccine coverage ($V_{80}$) | 0.93±0.026[3 11] |

HPV, human papillomavirus; LBC, liquid-based cytology.

uptake, with additional simulation results for 40% vaccine coverage given in online supplemental material.

## DISCUSSION

In all simulated cases with realistic parameters, HPV-reflex testing correctly identifies a higher proportion of cases ($97.1\% \pm 0.4\%$ to $99.6\% \pm 0.2\%$) relative to LBC testing for the same scenarios ($89.7\% \pm 0.4\%$ to $90.3\% \pm 0.1\%$). This is seen clearly in figure 3A,B, and detailed in table 2. The superior performance of HPV-reflex manifests chiefly due to markedly lower false positive rates. In all scenarios, LBC false positive rates exceed HPV-reflex values for all scenarios by a factor of 5.6–23.8. This analysis focused on a comparison of HPV reflex to LBC testing. It did not explicitly consider cotesting, where both LBC and HPV screening are performed and a positive on either leads to colposcopy, as this modality leads to unacceptably high false positives results, although it is possible in principle to extend the analysis here to consider how this scenario would vary by age cohort and vaccination status.

This difference is particularly pronounced in vaccinated cohorts, as false positives decline markedly with lower HPV prevalence in HPV-reflex testing. By contrast, false positive rates slightly increase in vaccinated cohorts with LBC testing, as seen in figure 3C,D. This is expected, as the limit of LBC false positive rate as CIN2+ prevalence approaches zero is simply $1 - s_{pl}$, yielding a theoretical maximum possible false positive rate for LBC of approximately 9.7%. By contrast, false positives decline with reflex with HPV, resulting in a theoretical minimum false

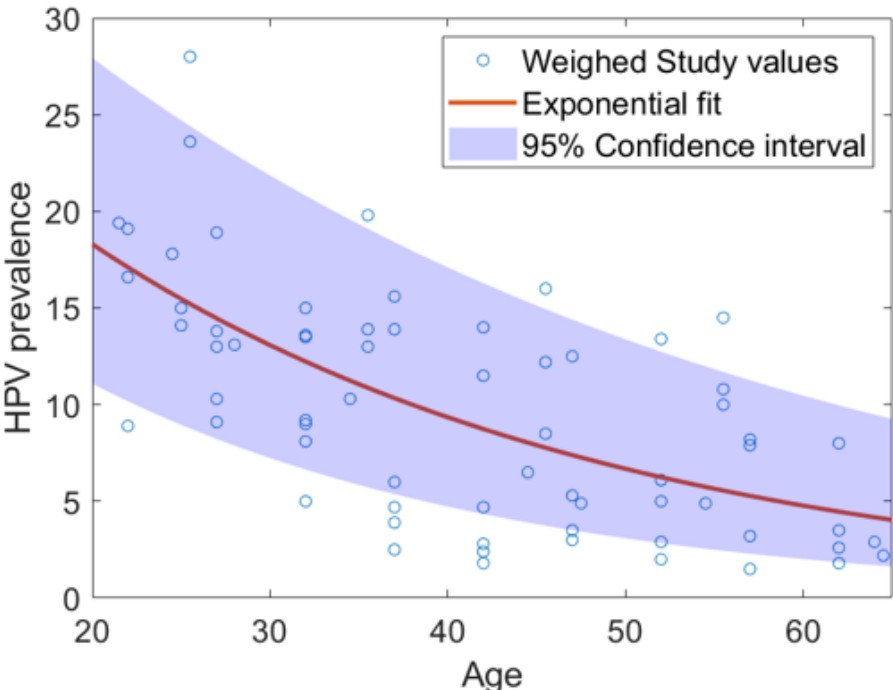

**Figure 2** Weighted fits from 12 European population studies with exponential fit and CIs. HPV, human papillomavirus.

**Table 2** Modelled test performance at specific ages and vaccination levels

| Age profile | LBC | | | HPV reflex | | |
|---|---|---|---|---|---|---|
| | False negative % | False positive % | Correct result % | False negative % | False positive % | Correct result % |
| Age 25 | | | | | | |
| 0% vaccinated | 0.8±0.28 | 9.38±0.14 | 89.1±0.31 | 1.05±0.26 | 1.46±0.27 | 97.50±0.40 |
| 40% vaccinated | 0.38±0.14 | 9.55±0.11 | 90.07±0.18 | 0.49±0.13 | 1.61±0.26 | 97.89±0.29 |
| 80% vaccinated | 0.06±0.03 | 9.68±0.10 | 90.27±0.11 | 0.07±0.03 | 1.73±0.26 | 98.20±0.26 |
| Age 35 | | | | | | |
| 0% vaccinated | 0.57±0.21 | 9.47±0.12 | 89.95±0.24 | 0.75±0.20 | 1.16±0.23 | 98.10±0.30 |
| 40% vaccinated | 0.27±0.10 | 9.59±0.11 | 90.14±0.15 | 0.35±0.10 | 1.26±0.22 | 98.38±0.24 |
| 80% vaccinated | 0.04±0.02 | 9.68±0.10 | 90.28±0.10 | 0.05±0.02 | 1.35±0.22 | 98.60±0.22 |
| Age 45 | | | | | | |
| 0% vaccinated | 0.41±0.16 | 9.54±0.12 | 90.05±0.20 | 0.53±0.15 | 0.94±0.19 | 98.53±0.24 |
| 40% vaccinated | 0.19±0.08 | 9.62±0.11 | 90.18±0.11 | 0.25±0.07 | 1.01±0.18 | 98.73±0.20 |
| 80% vaccinated | 0.03±0.02 | 9.69±0.10 | 90.28±0.10 | 0.04±0.01 | 1.07±0.18 | 98.89±0.18 |
| Age 55 | | | | | | |
| 0% vaccinated | 0.29±0.12 | 9.58±0.11 | 90.12±0.16 | 0.38±0.11 | 0.78±0.16 | 98.84±0.19 |
| 40% vaccinated | 0.14±0.06 | 9.65±0.10 | 90.22±0.12 | 0.18±0.06 | 0.84±0.15 | 98.99±0.16 |
| 80% vaccinated | 0.02±0.01 | 9.69± 0.10 | 90.29 ± 0.10 | 0.03 ± 0.01 | 0.88 ± 0.15 | 99.10 ± 0.10 |

HPV, human papillomavirus; LBC, liquid-based cytology.

positive rate of $(1 - s_{pl})(1 - s_{ph})$, or a minimum value of approximately 0.39%.

For both LBC and reflex testing, false negatives decline with decreasing HPV prevalence. LBC testing does have a slightly lower false negative rate than HPV-reflex testing, but this is marginal and overlaps with the performance envelope for HPV-reflex testing. In principle, LBC testing is more sensitive to instances of CIN2+ than reflex screening, but it results in far more false positives and unnecessary referrals to colposcopy. As HPV detection sensitivity is so high, expedited retesting can be employed when a patient tests positive for a high-risk strain of HPV and negative on LBC. This is likely to more than compensate for the marginally higher false negative rate of reflex testing, while preserving the crucial benefit and overall accuracy. Expedited retesting is already implemented for HPV positive results in several screening programmes, including the Irish National screening programme CervicalCheck.[27] where HPV positive but cytology-negative patients are recalled for repeat screening after 1 year rather than 3 or 5. Such approaches maintain the benefits of HPV reflex without the potential drawback of lower net sensitivity.

One limitation of the current study was the assumption that CIN2+ prevalence corresponds linearly to HPV prevalence, yielding an approximately constant proportion with age cohort. This is a reasonable assumption as justified in the mathematical appendix, but there are several important caveats that need to be considered. First, the proportion estimated in this work pivots on how well the populations in the HPV and CIN2+ datasets correspond to one another. While these datasets were chosen to avoid the potentially skewing influence of elective sexually transmitted disease clinic settings, there may be other confounding reasons why they are not representative, in which case estimates of $p(a)$ might be biased.

In addition, the proportionality itself might vary with time for biological reasons. While younger women, for example, might have a higher rate of HPV, it is conceivable that they have better mechanisms of clearance, and so proportionally might have a lower fraction of CIN2+ compared with to older cohorts. In the absence of clear data, this is of course speculative but the existence of any such age-specific variation in clearance rate would potentially impact the estimates in this work.

It is worth briefly discussing the mean HPV infection rate in unvaccinated populations across all age ranges, calculated as $\bar{h} = 9.4 \pm 2.3\%$ in this work. This figure is in good agreement with literature estimate of 8.4% as described, and with population-wide European estimates of 9.8%[25] (95% CI 9.2% to 10.0%) and more recent results from the ATHENA study.[26] Estimates in literature, however, can be someway complicated by how sampling conducted—samples drawn from opportunistic testing or from specific environments like sexual health clinics, or samples skewed heavily towards youth may tend to overestimate prevalence. Bruni et al,[25] for example, in their meta-analysis reported a prevalence of 14.2% (ranging from 8.8% in Southern Europe to 21.4% in Eastern Europe).

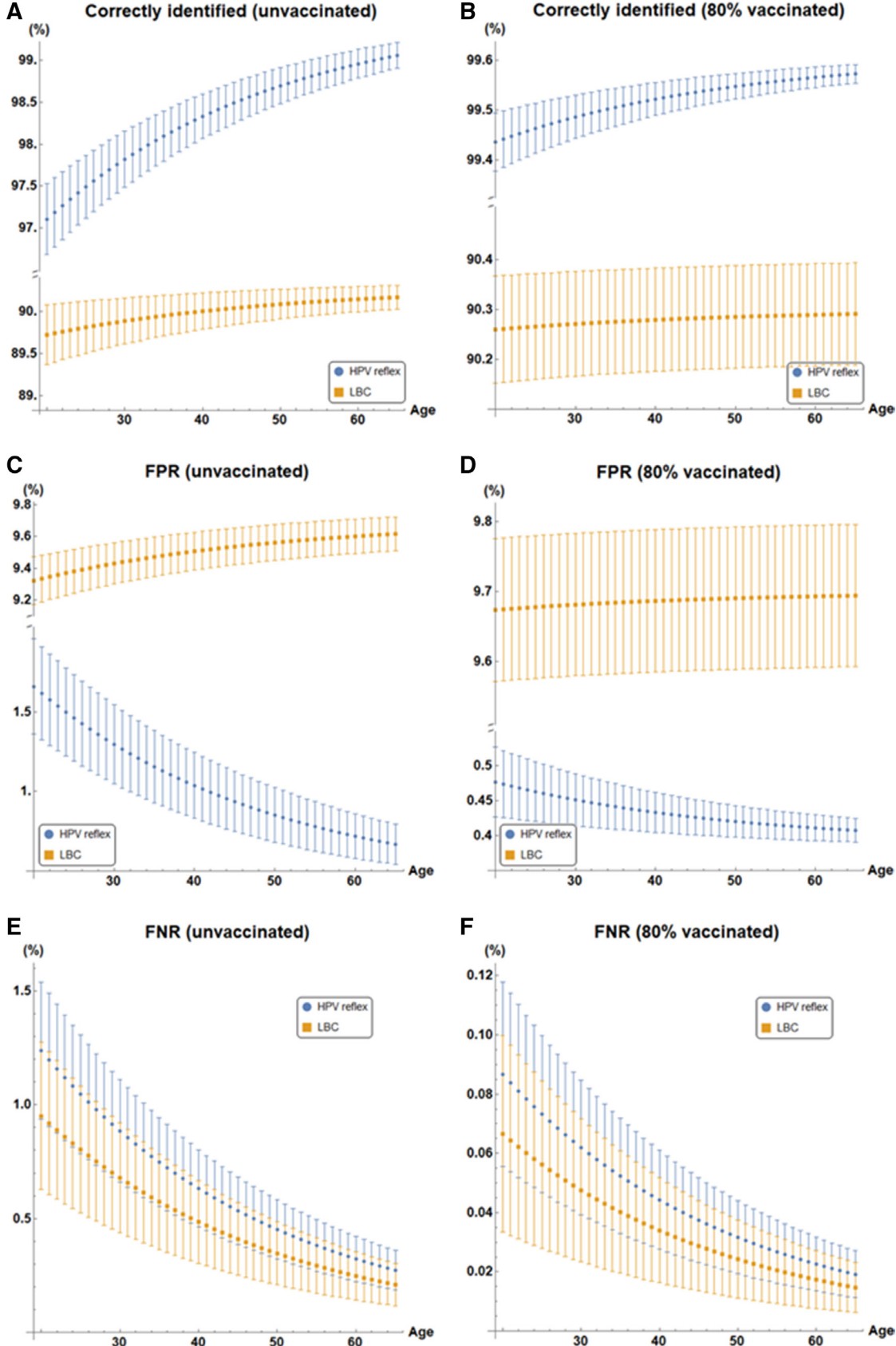

**Figure 3** Performance of HPV reflex and LBC screening modalities with age. Percentage correct results shown for (A) unvaccinated populations and (B) 80% vaccine uptake populations. False positive rates (FPR) of both modalities with age shown for (C) unvaccinated populations and (D) 80% vaccine uptake populations. False negative rates (FNR) depicted for (E) unvaccinated populations and (F) 80% vaccinated populations. Note the broken Y-axis in (A–D) inclusive. HPV, human papillomavirus; LBC, liquid-based cytology.

These estimates are readily confounded by the age and selection bias of subjects in the underlying studies (eg, studies disproportionately consisting of younger women or opportunistic testing), hence the disparity with the reported population prevalence even in the same work. In this model, $\bar{h}$ is data derived, but we can also force a higher value into the simulation to examine consistency ever with much higher prevalence. Online supplemental material contains a simulation assigning $\bar{h} = 15\%$ to investigate this and demonstrates that even this much higher contrived estimate has only negligible impact on false positive and negative rates.

It is important to note that this analysis did not consider the impacts of follow-up between screening rounds, nor the effects intervals between screening rounds, yielding instead an age-specific point estimate of false positive and negative rates in isolation. The cumulative false positive and negative rate over multiple screening rounds will vary depending on the intervals between screening, the rate of HPV clearance and the modalities employed. This is an involved question beyond the scope of this work, and the impacts of repeated screening on false negative and positive rates require dedicated future analysis to answer fully.

In consequence, HPV-reflex testing is calculated to be more robust to changing age and vaccine status than LBC testing and will increase further in accuracy as HPV infection decreases due to vaccination. This is an important consideration for the planning and interpretation of future screening and colposcopy services, which need to be robust for both age variation of HPV infection, and the eventuality where HPV infection is virtually eradicated due to increased uptake of the HPV vaccine. This is especially critical considering the World Health Organisations' cervical cancer elimination strategy 90-70-90 targets. These proposed measures,[28] to be put in place across nations worldwide by 2030 to eliminate cervical cancer, comprise 90% of girls vaccinated against HPV by age 15, 70% of women screened by age 35 and 90% of people identified with cervical disease treated. The results of this analysis strongly suggest that HPV-reflex screening approaches are robust to this changing epidemiology of HPV infection and will continue to perform strongly even as HPV infections fall with increasing vaccine uptake.

**Contributors** DRG conceived the model, analysed the published data, performed the analysis, reported the results, wrote the manuscript, and acts as guarantor for the content.

**Funding** This work was funded by grant number 214461/Z/18/Z from the Wellcome Trust.

**Disclaimer** The funder had no role in the design and conduct of the study; collection, management, analysis, and interpretation of the data; preparation, review, or approval of the manuscript; and decision to submit the manuscript for publication.

**Competing interests** DRG reported that he has written articles on both screening and HPV vaccination for popular audiences, receiving no industry funding for these articles. He also reports that he has consulted for public health bodies and charities in Europe on screening. He also reported receiving paid royalties for a popular science book and receiving fees and honoraria for contributed articles, fact-checking, public talks and media appearances.

**Patient and public involvement** Patients and/or the public were not involved in the design, or conduct, or reporting, or dissemination plans of this research.

**Patient consent for publication** Not applicable.

**Provenance and peer review** Not commissioned; externally peer reviewed.

**Data availability statement** Data sharing not applicable as no datasets generated and/or analysed for this study. Source code for the simulations herein are hosted online at https://github.com/drg85/CervScreenAgeVax.

**ORCID iD**
David Robert Grimes http://orcid.org/0000-0003-3140-3278

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
