## [Reviewer comments · BMJ Open]

ARTICLE DETAILS

TITLE (PROVISIONAL)	Impact of Human Papillomavirus age-related prevalence and vaccination levels on interpretation of cervical screening modalities – a modelling study
AUTHORS	Grimes, David

VERSION 1 – REVIEW

REVIEWER	Russell, Noirin University College Cork College of Medicine and Health, Obstetrics and Gynecology I am the Clinical Director for the Irish cervical screening programme
REVIEW RETURNED	05-Sep-2023

GENERAL COMMENTS	comment 1 I have concerns about the 8.4% prevalence quoted for HPV and refer the author to the paper by L. Bruni, M. Diaz, X. Castellsagué, E. Ferrer, F. X. Bosch and S. d. Sanjosé. Cervical Human Papillomavirus Prevalence in 5 Continents: Meta-Analysis of 1 Million Women with Normal Cytological Findings. The Journal of Infectious Diseases. 2010;202:1789 - 99 "In Europe, adjusted prevalence of HPV was estimated to be 14.2% (95% CI: 14.1 to 14.4).(135) Prevalence of HPV was highest in Eastern Europe (21.4%) and lowest in Southern Europe (8.8%).(135) Adjusted prevalence of HPV in Northern Europe, which included studies from Ireland, the UK, Sweden, Norway, Lithuania, Finland and Denmark, was 10.8% (95% CI: 9.8 to 10.2)." comment 2 I think this paper is important and contributes to the discussion about the future of cervical screening and the potential impact of HPV vaccination. The paper would benefit from adding some clinical commentary about the impact false positive screening results have on patients and on diagnostic services. Additionally, some commentary on the WHO Cervical Cancer elimination strategy would link this modelling paper into the clinical arena for further consideration. (The reviewer provided a marked copy with additional comments. Please contact the publisher for full details.)
---

REVIEWER	Vanska, Simopekka Finnish Institute for Health and Welfare
REVIEW RETURNED	17-Nov-2023

GENERAL COMMENTS	The author presents performance of different cervical cancer screening test modalities under vaccination, assuming a linear correspondence between HPV prevalence and severe disease stages (CIN2+). More precisely, the assumption is for the overall (any) HPV prevalence (line 44, page 5). As such, the assumption does not hold, because HPV types differ in their progression. This difference has been the rationale in selecting certain HPV types for vaccines. The elimination of the most progressing HPV types by vaccination changes the distribution of HPV types, reducing the potential of progression for the remaining HPV types. Thus, the vaccination also changes the correspondence between any HPV prevalence and severe disease stages, making the results fundamentally wrong.
---

VERSION 1 – AUTHOR RESPONSE

Responses to Dr. Noirin Russell

1. *I have concerns about the 8.4% prevalence quoted for HPV and refer the author to the paper by L. Bruni, M. Diaz, X. Castellsagué, E. Ferrer, F. X. Bosch and S. d. Sanjosé. (Cervical Human Papillomavirus Prevalence in 5 Continents: Meta-Analysis of 1 Million Women with Normal Cytological Findings. The Journal of Infectious Diseases, 2010;202:1789 – 99) - "In Europe, adjusted prevalence of HPV was estimated to be 14.2% (95% CI: 14.1 to 14.4).(135) Prevalence of HPV was highest in Eastern Europe (21.4%) and lowest in Southern Europe (8.8%).(135) Adjusted prevalence of HPV in Northern Europe, which included studies from Ireland, the UK, Sweden, Norway, Lithuania, Finland and Denmark, was 10.8% (95% CI: 9.8 to 10.2)."*

I thank Prof Russell for raising this point which requires some unpacking. Firstly in this work, \bar{h} refers to the HPV prevalence in the general population. The figure arrived at in this work is not an input parameter but derived from the data itself as outlined in the model. The seeming discrepancy with the figures Prof Russell cites have a nuanced explained; In Bruni et al, table 3 and figure 2 show the adjusted prevalence by age group and region, ranging from 24% for ages under 25 to 4.2% at ages 45-55. The prevalence estimate given here is skewed by over-representation of women in their 30s, who have a higher mean HPV prevalence (13.9%) than older age groups, and comprised the bulk of the sample. However, for our purposes we need a global population average, typically lower. In that paper, the authors write:

"Population-based studies showed an HPV prevalence of 9.8%, which was higher than that observed among women attending screening programs or participating in case-control studies", noting that for screening participants, the HPV prevalence was lower at 4.2%."

This population estimate of 9.8% is very close to the model prediction, which is reassuring. It is also close to the 2015 ATHENA study estimate for entire populations. I also took a look at the HIQA Health Technology assessment of HPV screening and I believe the higher figure for prevalence arises because the only Irish data available stemmed from opportunistic and non-systematic screening, which tends to inflate figures. For example, STI clinics with elective testing tend to have a prevalence well above population average. However, while \bar{h} arises from the model in this work, I also ran the simulation with an artificially high level of prevalence by forcing $\bar{h} = 15\%$. The reassuring thing is that the false positive

/ negative rates change only minimally, and this is outlined now in the supplementary material. Accordingly, to avoid confusion, text has been updated in several places:

Methods section:

“This figure pertains to the total prevalence across the population, and the caveats are elucidated in the discussion. “

Results section:

“Age-related HPV prevalence from 12 included studies were found to approximately obey an exponential decay type-model from ages 20-65 of the form $A \exp(-\beta t)$ as illustrated in figure 1 ($A = 35.81 \pm 4.91$, $\beta = 0.0336 \pm 0.0045 \text{ yr}^{-1}$, 95% confidence interval). This envelope with weighed study data points are shown in figure 1. The estimate for mean HPV infection rate in unvaccinated populations across all age ranges was $\bar{h} = 9.4 \pm 2.3 \%$, in good agreement with literature estimate of 8.4%, and with population wide European estimates of 9.8%²² (95% Confidence Interval: 9.2%-10.0%) and from the ATHENA study²³ (10.5%, 95% Confidence Interval: 10.3%-10.7%) . CIN2+ rates by age cohort from inferred from this as per the model outline. “

Discussion section:

“It is worth briefly discussing the mean HPV infection rate in unvaccinated populations across all age ranges, calculated as $\bar{h} = 9.4 \pm 2.3 \%$ in this work. This figure is in good agreement with literature estimate of 8.4% as described, and with population wide European estimates of 9.8%²² (95% Confidence Interval: 9.2%-10.0%) and more recent results from the ATHENA study²³. Estimates in literature however can be somewhat complicated by how sampling conducted – samples drawn from opportunistic testing or from specific environments like sexual health clinics, or samples skewed heavily towards youth may tend to over-estimate prevalence. Bruni et al², for example, in their meta-analysis reported a prevalence of 14.2% (ranging from 8.8% in Southern Europe to 21.4% in Eastern Europe.

These estimates are readily confounded by the age and selection bias of subjects in the underlying studies (for example, studies disproportionately consisting of younger women or opportunistic testing), hence the disparity with the reported population prevalence even in the same work. In this model, \bar{h} is data derived, but we can also force a higher value into the simulation to examine consistency even with much higher prevalence. The supplementary material contains a simulation assigning $\bar{h} = 15\%$ to investigate this incidence, and demonstrates that even this much higher contrived estimate has only negligible impact on false positive and negative rates. “

Supplementary material – New tables for $\bar{h} = 15\%$

- 2. I think this paper is important and contributes to the discussion about the future of cervical screening and the potential impact of HPV vaccination. The paper would benefit from adding some clinical commentary about the impact false positive screening results have on patients and on diagnostic services. Additionally, some commentary on the WHO Cervical Cancer elimination strategy would link this modelling paper into the clinical arena for further consideration.*

I thank Prof Russell for her considered and thoughtful remarks, and agree fully. I have addressed several comments in the appended marked PDF, and included reference to the WHO CC elimination strategy. For brevity, I have marked changes in the attached document.

Responses to Dr. Simopekka Vanska

- 1. The author presents performance of different cervical cancer screening test modalities under vaccination, assuming a linear correspondence between HPV prevalence and severe disease stages (CIN2+). More precisely, the assumption is for the overall (any) HPV prevalence (line*

44, page 5). As such, the assumption does not hold, because HPV types differ in their progression. This difference has been the rationale in selecting certain HPV types for vaccines

I thank Dr Vanska for this comment, and I would like to remedy any misconceptions I may have inadvertently introduced with a clarification and mathematical argument. Firstly, we can show that CIN2+ prevalence will be directly proportional to HPV prevalence, even accounting for strain types and their respective likelihood of causing lesions. Consider n strains of HPV, each constituting a proportion p of the total HPV burden. The HPV burden may be written as

$$H = p_1 + p_2 \dots + p_n = \sum_1^n p_n.$$

We may write p as a vector, and defining u as a vector of length n with every entry equal to unity, we note through the properties of the dot product operator that it follows that

$$H = p \cdot u = \sqrt{n}|p| \cos \phi \rightarrow |p| = \frac{H}{\sqrt{n} \cos \phi}$$

where ϕ is the angle between the vectors. We can further state that each strain of HPV has a probability of becoming a CIN2+ lesion of c_i respectively per strain, ranging from 0 to 1. Thus the total lesion burden is given by

$$C = c_1 p_1 + c_2 p_2 \dots + c_n p_n = \sum_1^n c_n p_n.$$

The values for p and c can be vectors and from the properties of the dot product, we may write

$$C = p \cdot c = |p||c| \cos \theta = \frac{H |c| \cos \theta}{\sqrt{n} \cos \phi}.$$

where θ is the angle between the prevalence and risk vectors. As can be seen here, it follows that no matter the configuration of these vectors, $C \propto H$ and so the assumption is always justified, regardless of the individual strains involved. In the current document, the above mathematical argument has been added to the appendix, and the methodological text amended to allude to the mathematical justification.

2. *The elimination of the most progressing HPV types by vaccination changes the distribution of HPV types, reducing the potential of progression for the remaining HPV types. Thus, the vaccination also changes the correspondence between any HPV prevalence and severe disease stages, making the results fundamentally wrong.*

It is fundamentally true that vaccination efficacy differs between strains, and long-term elimination of various strains is best modelled by natural history approaches. This is not the intention of this work, which simply wishes to assess how screening approaches would cope under various scenarios. The cited figure on HPV prevalence after vaccination is empirical, derived from previous modelling studies, and functions simple as a multiplier to gauge the performance of HPV-reflex screening under reduced incidence of infection, independent of strain. We can however use a similar argument to the previous response to show here why the approximately linear correspondence holds. Let v be the vector efficacy of vaccination for each strain, with each entry given by

$$v_i = (1 - e_i)$$

where e_i is the fractional respective efficacy of the vaccine against each strain, bounded between 0 and unity. We denote an element-by-element Hadamard product, corresponding to the reduced prevalence

of each HPV subtype after vaccination as $p_v = p \odot v$, and like before, the modified HPV prevalence after vaccination is given by $H_v = p_v \cdot u$. But as the Hadamard product is commutative, we can instead write this as $H_v = p \cdot (u \odot v)$. Letting $u \odot v = u_v$ for brevity, then

$$H_v = p \cdot u_v = \sqrt{u_v} |p| \cos \phi_{uv} \rightarrow |p| = \frac{H_v}{\sqrt{u_v} \cos \phi_{uv}}.$$

This can be set equal to the value for $|p|$ previously derived and rearranged to establish the identity

$$H_v = H \left(\frac{\sqrt{u_v} \cos \phi_{uv}}{\sqrt{n} \cos \phi} \right).$$

The bracketed term is effectively a constant, corresponding to the reduction in net HPV infection. This is akin to the $(1 - V)$ term in the work itself, demonstrating the empirical use of a linear reduction factor derived from previous studies is justified to ascertain the performance of screening modalities under different levels of coverage. This argument has now been added to the supplementary material, and alluded to in the main text.

VERSION 2 – REVIEW

REVIEWER	Russell, Noirin University College Cork College of Medicine and Health, Obstetrics and Gynecology
REVIEW RETURNED	05-Jan-2024
GENERAL COMMENTS	The revisions have improved the paper clarity. I have suggested adding two references -one to support anxiety in patients referred to colposcopy and one to support potential increasing referrals to colposcopy. References will underline that these are real concerns. please insert after sentence below False positive results not only induce anxiety in patients but also lead to potential overtreatment, with the potential to overwhelm colposcopy services unless these are robustly designed.

VERSION 2 – AUTHOR RESPONSE

I concur with Dr Russell that references to the anxiety and overwhelming of colposcopy should be included. I have now added three relevant references. Please let me know if this suffices, and if I can provide anything else.